# High-Resolution Imaging of Cones and Retinal Arteries in Patients with Diabetes Mellitus Type 1 Using Adaptive Optics (rtx1)

**DOI:** 10.3390/biomedicines12040863

**Published:** 2024-04-14

**Authors:** Wojciech Matuszewski, Michał Szklarz, Katarzyna Wołos-Kłosowicz, Joanna Maria Harazny, Elżbieta Bandurska-Stankiewicz

**Affiliations:** 1Clinic of Endocrinology, Diabetology and Internal Medicine, Department of Internal Medicine, University of Warmia and Mazury, 10−082 Olsztyn, Poland; 2. Department of Human Physiology and Pathophysiology, University of Warmia and Mazury, 10−082 Olsztyn, Poland; joanna.harazna@uwm.edu.pl; 3Department of Nephrology and Hypertension, University Hospital Erlangen, Friedrich Alexander University Erlangen Nuremberg (FAU), 91054 Erlangen, Germany

**Keywords:** diabetes mellitus type 1, diabetic retinopathy, adaptive optics, cones

## Abstract

(1) Background. Diabetes mellitus (DM), called the first non-infectious epidemic of the modern era, has long-term health consequences leading to a reduced quality of life, long-term disabilities, and high mortality. Diabetic retinopathy (DR) is a neurovascular complication of diabetes and accounts for about 80% cases of vision loss in the diabetic population. The adaptive optics (AO) technique allows for a non-invasive in vivo assessment of retinal cones. Changes in number or morphology of retinal cones may be one of the first indicators of DR. (2) Methods. This study included 68 DM1 patients (17 women) aged 42.11 ± 9.69 years with a mean duration of diabetes of 22.07 ± 10.28 years, and 41 healthy volunteers (20 women) aged 41.02 ± 9.84 years. Blood pressure, BMI, waist circumference, and metabolic control measures were analysed. Cones’ morphological parameters were examined with a retinal camera with Imagine Eyes adaptive optics (rtx1). Statistical analysis was carried out with IMB SPSS version 23 software. (3) Results. Neither study group differed significantly in age, BMI, blood pressure, or eyeball length. Intraocular pressure (IOP) was statistically significantly higher in DM1 patients but remained within physiological range in both groups. Analysis of cone parameters showed a statistically significant lower mean regularity of cones (Rmean) in the DM1 group compared to control group (*p* = 0.01), with the lowest value in the group with DM1 and hypertension (*p* = 0.014). In addition, DM1 patients tended to have fewer cones. (4) Conclusions. Our study revealed abnormalities in cone and vessel parameters and these abnormalities should be considered as risk factors for the development of DR. Complementing an eye examination with AO facilitates non-invasive in vivo cellular imaging of the retina. Lesions like those detected in the eye may occur in the brain and certainly require further investigation.

## 1. Introduction

Diabetes mellitus (DM), called the first non-infectious epidemic of the modern era, has a number of long-term health consequences, affecting patients’ well-being and burdening the healthcare system. The disease leads to numerous disabilities and causes high mortality due to its multiple chronic vascular complications [1,2]. Currently, according to the International Diabetes Federation (IDF), more than 537 million people have diabetes, and it is estimated that by 2045, this number will increase to 783 million people worldwide. In Europe alone, there are 61 million people currently diagnosed with diabetes in Europe, and by 2045, a 13% increase is estimated, resulting in 69 million patients [3]. Type 1 diabetes mellitus (DM1) accounts for 15–20% of all cases of the disease, and is particularly common in European populations [4,5]. Diabetic retinopathy (DR) is classified as a neurovascular complication and it is the most serious ocular complication caused by diabetes. It might already be present at the time of diabetes diagnosis, and accounts for approximately 80% of cases of vision loss in the diabetic population [6,7]. A detailed meta-analysis of 59 population-based studies published in 2021 confirmed that the global prevalence of DR in diabetic population was 22.27%. In 2020, the number of adults with DR was 103.12 million, including 28.54 million with vision-threatening DR (VTDR) and 18.83 million with diabetic macular oedema (DMO) worldwide. Predictions for 2045 are unfavourable, forecasting an increase to 160.50 million, 44.82 million, and 28.61 million, respectively [8]. Adaptive optics (AO) is a technique first introduced by David Williams from Rochester, USA and it is still one of the most accurate ways of assessing the vasculature and photoreceptor structure of the retina [9]. Cones are responsible for seeing colours, shapes, and adequate visual acuity in bright light, i.e., photopic vision. There are three types of cones: short, medium, and long, responding to light from a different wavelength range, 420 nm, 534 nm, and 564 nm, respectively [10,11]. Pathophysiological mechanisms underlying the development of DR include neurovascular damage to the retina, thus creating a disadvantageous metabolic environment for cones [12,13]. Changes in the number or morphology of cones may be one of the first signs of DR. Considering that DR is a separate risk factor for myocardial infarction or stroke, the use of AO is of key importance, providing thorough analysis of structural and functional changes of the retina [14,15,16,17]. Hence, the aim of our study was to evaluate the morphology and function of retinal cones in DM1 patients without previously diagnosed DR compared to a group of healthy volunteers.

## 2. Material and Methods

### 2.1. Study Population

The study was conducted in 2021–2022 according to the Good Clinical Practice guidelines and included patients of the Diabetes Outpatient Clinic at the Department of Endocrinology, Diabetology and Internal Medicine at the University of Warmia and Mazury in Olsztyn, Poland. A total of 68 adults, including 27 DM1 patients (17 female) without DR, aged 42.11 ± 9.69 years (median 40 IQR 37–50 years), and 41 healthy volunteers (20 female) aged 41.02 ± 9.84 years (median 42 IQR 34–48 years) were enrolled in the study. Patient population consisted of adults diagnosed with DM1 according to WHO criteria, with no features of DR, treated with functional intensive insulin therapy according to the American Diabetes Association (ADA) and European Association for the Study of Diabetes (EASD) guidelines [4,18,19,20]. All patients provided informed consent prior to the enrolment. The study was approved (approval number 10/2010 of 25 March 2010) by the Bioethics Committee of the Faculty of Medical Sciences of the University of Warmia and Mazury in Olsztyn, Poland.

### 2.2. Methods of Metabolic Compensation Assessment

Metabolic compensation was analysed across the entire study population using self-reported questionnaires created for the purpose of the study, which included the following: anthropometric data, clinical history, and elements of physical examination. Hypertension was defined according to the European Society of Hypertension (ESH) and European Society of Cardiology (ESC) criteria; BMI was calculated by Quetelet’s formula [21,22]. Laboratory parameters were also assessed: glycated haemoglobin (HbA1c), lipid profile, creatinine levels, glomerular filtration rate (GFR), and urine albumin-creatinine ratio (UACR) (Table 1).

### 2.3. The Ocular Assessment Methods

Funduscopic examination was performed with a Topcon TRC NW8 fundus camera after pupil dilation with 1% Tropicamide. An experienced ophthalmologist took colour two-field fundus photographs of both eyes (with a 50-degree angle), encompassing the optic nerve disc in the centre and the macula of the retina in the centre. Results were assessed according to the International Clinical Diabetic Retinopathy criteria [18]. Examination of the eyeball length was performed with an IOL Master 500 optical biometer (Zeiss, Jena, Germany), and intraocular pressure was measured with an Air-Puff TX-20 type non-contact tonometer (Canon, Tokyo, Japan). The retina was examined with an adoptive optics rtx1 camera (Imagine Eyes, Orsay, France), following the manufacturer’s protocol. The retina was examined in vivo at the cellular and microvascular level. The following parameters were measured and calculated automatically: vessel wall thickness (wall 1 + wall 2) (WT), lumen diameter (LD), wall-to-lumen ratio (WLR), and vascular wall cross-sectional area (WCSA). All measurements of retinal arterioles were taken in the superior temporal quadrant (before the second juxtapapapillary bifurcation) in triplicate, and the arithmetic mean of the measurements was calculated. Individual eyeball lengths were taken into account in the analysis and left and right eye test results were averaged. Patients were examined by a trained ophthalmologist with more than 25 years of research experience in retinal microfluidics and carotid artery morphology. Prior to the examination, patients were allowed 10 min of rest in a sitting position, in the dark, with no pupil dilation, in an air-conditioned room (23 °C). The observer was blinded to the participant’s group. Blood pressure tests were performed with an Omron M3 blood pressure monitor (Omron, Kyoto, Japan). Central blood pressure and heart rate were measured with a SphygmoCor Xcel PWA/PWV (AtCore, Syndney, Australia). Morphological parameters of retinal cones were measured with the AO rtx1 camera in right and left eye. The perifoveal region with the highest quality and sharpness of cone visualisation was selected for the analysis. The following parameters were assessed: *n*—number of cones measured in the perifoveal region in the right and left eye; D—density of cones per square millimetre of the retinal area measured in the perifoveal region in the right and left eye; S—cone spacing measured in the perifoveal region in the right and left eye, and spatial distribution of cones was analysed in terms of inter-cones spacing; R—cone regularity measured in the perifoveal region in the right and left eye, i.e., the sum of the percentage of pentagonal (*n*% 5), hexagonal (*n*% 6), and heptagonal (*n*% 7) cones; DP—cone dispersion index measured in the perifoveal region in the right and left eye. Carotid vessels were assessed with ultrasound (Samsung, Korea) of the common carotid artery in diastole, 2 cm before bifurcation into the internal/external carotid arteries, in a sitting position, after 5 min of rest; results for the right and left eyes were averaged. Additionally, the following parameters were measured: IM—intima/media thickness of the left and right common carotid artery; LDCA—lumen diameter of both of the common carotid arteries; IMTLR—intima/media thickness lumen ratio calculated with IMT/LDCA in both common carotid arteries.

### 2.4. Statistical Analyses

Statistical analyses were performed using version 28 of IMB SPSS programme (IBM New York, NY, USA). The *p*-value for differences in median values for measurements in the absence of a normal distribution of results in at least one of the studied groups was compared using the non-parametric Mann–Whitney U test and the Kolmogorov–Smirnov test. Spearman’s non-parametric correlations of tested parameters were determined, adjusted additionally for age, gender, and BMI. The result of the analyses was considered statistically significant at *p* < 0.05.

## 3. Results

The two groups differed only in HDL cholesterol levels and HbA1c percentage, values of which were statistically significantly higher in the DM1 group (Table 1), with the remaining clinical parameters not significantly different between the two groups. A more detailed examination of lipid profiles of both groups in univariate linear analysis with age, gender, and BMI as co-variates showed statistically non-significantly higher HDL and LDL cholesterol values in the DM1 group (*p* = 0.083) (Table 2).

In the DM1 group there were no statistical differences in the eyeball lengths of both eyes: OD 23.66/OS 23.64 mm vs. OD 23.56 mm/OS = 23.61 mm as compared to the control group. There were no differences between groups in measured pressures, only the intraocular pressure was statistically significantly higher in the DM1 group yet remained within physiological range in both groups (Table 3).

Morphological analysis of the retinal vasculature revealed statistically significantly higher WT, WLR, and WCSA in the DM1 group compared to the control group (Table 4).

Morphological analysis of cone parameters showed statistically significantly lower cone regularity in the DM1 group compared to the control group. There was also a trend towards a reduced number of cones in the DM1 group (Table 5).

Morphological analysis of the common carotid arteries showed statistically significantly higher IMT and IMTLR values in the DM1 group compared to the control group. There were no differences in the lumen diameter between the DM1 group and the control group, similarly to the results of this parameter for retinal arterioles (Table 6).

Analysis of HbA1c values controlled by age, gender, and BMI in the DM1 showed a statistically significant correlation with IMT (*p* = 0.005) and IMTLR (*p* = 0.003). Analysis of DM duration values controlled by age, sex, and BMI showed statistically significant correlations with PPb (*p* = 0.021), WLR (*p* = 0.027), and IMTL R (*p* = 0.01). Analysis of WLR values controlled by age, sex, and BMI in the DM1 group showed a statistically significant correlation with DM duration (*p* = 0.027), SBPc (*p* = 0.045) and a high correlation with SBPb (*p* = 0.059).

Hypertension in both groups increased the WLR, which was significantly higher in the DM1 group as compared to controls. IMTLR mean was significantly higher in the DM1 group compared to the control group in patients both with and without hypertension (Figure 1).

In the DM1 group with hypertension, the number of cones was significantly lower as compared to the DM1 group without hypertension (*p* = 0.016) and control group with hypertension (*p* = 0.05). In the control group, there was no statistical difference in the number of cones in participants with and without hypertension. R mean was significantly lower in the DM1 group compared to the control group (*p* = 0.018) and reached its lowest value in the DM1 group with hypertension (*p* = 0.014) (Figure 2).

In both groups, there was a significant negative correlation between number of cones and peripheral and central blood pressure values: peripheral systolic blood pressure SBPb (r = −0. 34, *p* = 0.007), peripheral diastolic DBPb (r = −0.32, *p* = 0.010), central systolic blood pressure SBPc (r = −0.31, *p* = 0.013), and central diastolic blood pressure DBPc (r = −0.30, *p* = 0.015). No correlation was found with other vascular parameters, HBA1c or diabetes duration. However, R mean in all participants correlated significantly (r = −0.281; *p* = 0.026) with diabetes duration controlled by age, gender, and BMI, in Spearman’s correlation (r = −0.302, *p* = 0.014).

## 4. Discussion

The human retina contains approximately 4.6 million cones (4.08–5.29 million). The maximum cone density in the fovea averages 199,000 cones/mm^2^, but individual variations can range from 100,000 to as many as 324,000 cones/mm^2^. Hence, while studying any results obtained with AO, it is important to recall pioneering research in this field. One of the earliest studies based on AO assessment, published in 1990 by Curcio et al., showed average densities of 37,000 cones/mm^2^ at distances of 0.5 mm from the central fovea to 16,000 cones/mm^2^ at distances of 1.0 mm from the central fovea in a post-mortem study on seven individuals aged 27–44 years [23,24]. Park et al. focused on the importance of the study design and location of the photoreceptors studied. They evaluated 192 healthy volunteers (45.7% female) with a mean age of 33.6 years (SD 13.2) and a mean axial length of 24.4 mm (SD 1.41). The ethnic structure of the studied group varied, with people of Asian origin constituting 25.5%, African—11.5%, Caucasian—35.4%, and Hispanic—27.6%. Park showed different cone packing density depending on changes in retinal eccentricity. He demonstrated a gradual decrease in cone density from the fovea towards the periphery in all subjects along all meridians. The cited study demonstrated that average cone packing densities at 0.5, 1.0, and 1.5 mm from the centre of the fovea were approximately 32,199, 19,328, and 11,597 cones/mm^2^ [25]. In another study, 20 healthy volunteers divided into age groups were examined with AO: 10 younger ones aged 22–25 years, and 10 older ones aged 50–65 years. In the entire group, the mean cone density was 37,000 cones/mm^2^ at 0.5 mm and 19,000 cones/mm^2^ at 1.1 mm eccentricity, respectively. In our study, this distance was 0.3 mm. The greatest difference between the groups was noted at a distance of 0.18 mm from the foveal centre, while at a distance of 0.5 mm similar cone densities were found in both groups [26]. In 2010, Ooto et al. also examined 20 healthy volunteers, showing an average cone density of 33,320 and 14,450 cones/mm^2^ at distances of 0.5 mm and 1.0 mm from the central fovea. The second group consisted of patients with central serous chorioretinopathy (CSC), who showed significantly lower average cone density at each distance from the central fovea (0.5 mm *p* = 0.007; 1.0 mm *p* = 0.004). These results point to other ocular conditions that may affect measurements taken with AO [27].

In 2016, Lombardo et al. presented an interesting study, similar to ours, but with a smaller number of DM1 patients. In that study, 16 patients were enrolled. Half of them had no DR features, their mean age was 37.0 ± 6.6 years, with diabetes duration of 10.5 ± 2 years and HbA1c 7.5 ± 0.8%. The remaining half of patients had mild non-proliferative DR (NPDR), their mean age was 42.8 ± 9.0 years, with diabetes duration of 17.9 ± 8.5 years and HbA1c 7.3 ± 0.7%. The control group consisted of 20 healthy participants aged 36.4 ± 8.6 years. The average cone density was statistically significantly lower in the NPDR group compared to the no-DR group (26,585 ± 1377 cones/mm^2^ vs. 27,855 ± 970 cones/mm^2^, *p* < 0.001), and similarly to our results, it was the highest in the control group (29,452 ± 1484 cones/mm^2^). The linear dispersion index (LDi) was the highest in the NPDR group compared to no-DR and control groups, respectively: 0.088 ± 0.006; 0.085 ± 0.004; 0.077 ± 0.004, *p* < 0.001. The heterogeneity packing index was lower in the NPDR group compared to no-DR, and the highest in the control group, respectively: 0.369 ± 0.029; 0.375 ± 0.032; 0.431 ± 0.024, *p* < 0.001. Regarding cone parameters, i.e., average cone density LDi, the heterogeneity packing index showed a significant negative correlation with diabetes duration (*p* < 0.001). In our study, the R mean in all participants correlated significantly negatively with diabetes duration adjusted by age, gender, and BMI (*p* = 0.014) [28]. Another report concerning DM1 patients comes from a Romanian study involving 15 patients with DM1 without features of DR, aged 36.4 ± 6.46 years, with a mean diabetes duration of 19.13 ± 7.47 years. The control group consisted of 16 healthy participants aged 39 ± 7.75 years. As in our study, statistically significantly lower average cone density was found in all measured retinal areas in the DM1 group compared to the control group. The highest statistical significance was observed in the nasal retinal localisation: 22,782 ± 3677 cones/mm^2^ in the DM1 group and 25,725 ± 3815 cones/mm^2^ in the control group (*p* < 0.001). There was no correlation of cone density with age, sex, and diabetes duration [29]. A study of DM1 patients using AO was also conducted by Lammer et al., who examined 53 patients with a mean age of 44 ± 12.8 years, including 31 with DM1 with mean HbA1c of 8.1 ± 1.7%, and 12 with DM2 with mean HbA1c of 7.8 ± 1.8%. The age structure and poor metabolic compensation point to the similarities between these patients and those in our study, but Lammer et al. assessed patients who had already exhibited ocular complications. Among the DM patients, 26% had no features of DR, 62% had NPDR and 11% had proliferative DR (PDR), while 21% showed diabetic macular oedema (DMO). Nevertheless, contrary to the studies cited above and our results, analyses of cone parameters (cone density, cone spacing) showed no significant differences between the groups with and without DM. Moreover, no differences in terms of cone number were shown between different stages of DR advancement. Only patients with DMO exhibited significantly decreased cone density compared to patients without DMO (*p* = 0.04). Compared to healthy participants, DM patients showed decreased regularity of cone arrangement in the macular quadrants, as well as in different DR stages and in the presence of DMO (*p* = 0.04). Similarly to the present study, no changes in cone parameters were shown in relation to metabolic compensation of diabetes [30]. Other interesting study was published by researchers from Toronto. Tan et al., using the AO technique, evaluated 29 DM1 patients aged 19.06 ± 3.0 years with diabetes duration of 10.69 ± 3.9 years and mean HbA1c of 8.5 ± 1.3%. Examined patients were much younger compared to our study group (42.11 ± 9.69 years) but they all had in common poor metabolic control of diabetes. The control group consisted of 44 healthy volunteers aged 18.51 ± 3.36 years. In contrast to our study, the results showed no significant differences in cone density between the groups (DM1 mean = 10,298 ± 1635 cones/deg; control mean = 10,224 ± 1631 cones/deg; *p* = 0.60). The researchers explained that the result may have been affected by the young age of the participants [31]. The age aspect was also addressed by Elsner, who published a study based on AO-derived results in March 2022. 10 DM patients with a mean age of 54.7 ± 12.8 years were studied and compared to a younger (*n* = 26) and older (*n* = 10) control group of healthy volunteers, aged 24.4 ± 3.42 and 56.3 ± 3.71 years, respectively. A higher total number of cones (total cones) was observed in the younger control group compared to the older one: 238,000 ± 18,300 vs. 214,000 ± 33,000. However, the lowest number of cones was recorded in the DM group compared to both control groups of healthy volunteers, which confirms our results [13]. Another study assessed 25 DM2 patients, among whom 6 patients had mild NPDR, 7 moderate NPDR, 3 severe NPDR, 2 PDR, while 7 had no DR features. The control group consisted of 10 healthy volunteers. The cone density was significantly lower in the moderate NPDR, severe NPDR and PDR groups compared to the control, no-DR, and mild NPDR groups (*p* < 0.05). There was no correlation between cone density and HbA1c or diabetes duration, which in our study showed a significant negative correlation with cone regularity. Furthermore, having measured cone spacing, statistical significance was noted between no-DR and severe NPDR-PDR groups (*p* < 0.01) and between control and moderate NPDR-PDR groups (*p* < 0.003) [32].

It is also important to mention the results of Szaflik’s study, which were obtained using AO rtx1 assessing diabetic patients with DR: 22 with mild and 14 with moderate NPDR, compared to 20 healthy volunteers. Inclusion criteria encompassed balanced blood glucose and blood pressure values, and BMI lower than 25 kg/m^2^. Similarly, our study did not include participants with BMI > 25 kg/m^2^. In Szaflik’s study, the majority of patients received insulin (89%), only four were treated with oral antihyperglycemic drugs. Significant differences were found in the average cone density in the control and DR groups: 24,722 ± 3507 cones/mm^2^ and 19,822 ± 4342 cones/mm^2^, respectively (*p* < 0.001). The cone density decreased as DR progressed 20,440 ± 4522 cones/mm^2^ in mild NPDR, 18 688 ± 3 919 cones/mm2 in moderate NPDR (*p* 0.26). The interphotoreceptor spacing was significantly higher in the DR group compared to the control group. The mean percentage of cones with hexagonal Voronoi tiles (*n*% 6) in the control and DR groups was 47.7 ± 5.9% and 42.1 ± 4.4%, respectively (*p* < 0.001). The mean percentage of hexagonal cones was significantly lower in both DR subgroups, but there were no significant differences between subgroups (*p* = 0.505). The measured cone parameters did not depend on BMI. In contrast to our study, there was no correlation between cone and artery parameters and any of the blood parameters (*p* > 0.05). However, as in this study, statistically significant differences were observed between the DM1 and control groups in the vascular parameters WLR and WCSA (*p* < 0.001) [33].

## 5. Conclusions

Our study revealed abnormalities in cone and vessel parameters, which should be considered as early risk indicators for the development of DR. As hypertension may accelerate this process, in addition to strict metabolic control, patients should be also examined for hypertension, and if hypertension was diagnosed, its treatment should be implemented immediately to prevent or delay DR onset.

AO retinal evaluation aligns with the definition of DR as a neurovascular complication of diabetes. Complementing an eye examination with AO facilitates non-invasive in vivo cellular imaging of the retina. Lesions similar to those detected in the eye may occur in the brain and certainly require further investigation.

## Figures and Tables

**Figure 1 biomedicines-12-00863-f001:**
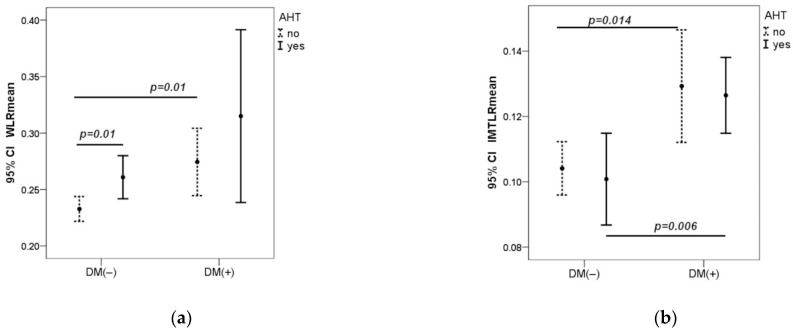
Dependence of WLR and IMTLR mean on arterial pressure in both groups (**a**,**b**). AHT-arterial hypertension, measured in the brachial artery. AHT (no) if SBPb < 140 mmHg and DBPb < 90 mmHg; AHT (yes) if SBPb ≥ 140 mmHg or DBPb ≥ 90 mmHg. CI—Confidence Interval. *p*—value was calculated by Mann–Whitney U test.

**Figure 2 biomedicines-12-00863-f002:**
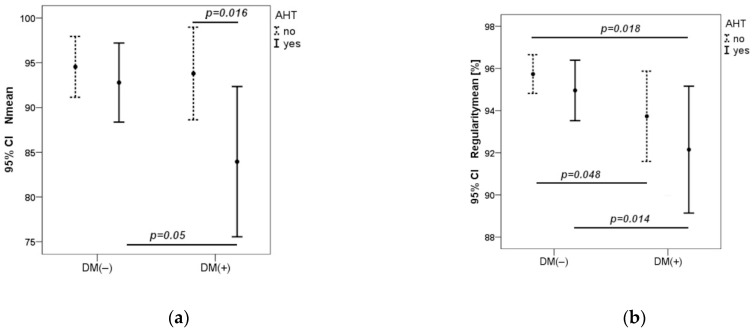
Dependence of cone number and regularity on blood pressure in both groups (**a**,**b**). CI—Confidence Interval. *p*—value was calculated by Mann–Whitney U test.

**Table 1 biomedicines-12-00863-t001:** Normal and recommended ranges of the assessed markers of metabolic control of diabetes.

Assessed Laboratory Indicators	Recommended Range
Glycated hemoglobin (HbA1c)	≤7.0% (≤53 mmol/mol)
Concentration of total cholesterol	<200 mg/dL (<5.2 mmol/L)
Concentration of HDL	>40 mg/dL (>1.0 mmol/L) in men >45 mg/dL (>1.2 mmol/L) in women
Concentration of LDL	<100 mg/dL (2.6 mmol/L)
Concentration of triglycerydes	<150 mg/dL (<1.7 mmol/L)
Concentration of creatinine	0.6–1.3 mg/dL (53–115 µmol/L)
Glomerular filtration rate (GFR)	<90 mL/min/1.73 m^2^
albumin/creatinine ratio (ACR)	<2.5 mg/g creatinine

**Table 2 biomedicines-12-00863-t002:** Characteristics of the analysed group with metabolic control markers. BMI—body mass index. SD—standard deviation; IQR—interquartile range between 25th percentile and 75th percentile.

Clinical Parameter	Control Group		DM1		*p* Mann–Whitney U test
	Mean ± SD	Median IQR	Mean ± SD	Median IQR	
*n* (%)	41 (60)	-	27 (40)	-	
Age—years (SD)	41.02 ± 9.84	40 37–50	42.11 ± 9.69	42 34–48	0.75
Diabetes duration [years]	-	-	22.07 ± 10.28	24 15–29	-
HbA1c [%]	5.23 ± 0.20	5.20 5.10–5.48	7.58 ± 1.14	7.45 6.78–8.68	<0.001
BMI [kg/m^2^]	24.6 ± 3.6	24.5 21.7–26.2	24.2 ± 2.6	23.6 22.0–26.2	0.88
Total Cholesterol [mg/dL]	179.7 ± 24.2	179.5 160.0–193.3	196.5 ± 46.3	190.5 157.5–232.0	0.23
HDL cholesterol [mg/dL]	64.6 ± 14.6	61.0 54.0–76.5	77.0 ± 21.0	72.5 63.0–93.5	0.049
LDL cholesterol [mg/dL]	99.7 ± 41.1	108.0 67.0–118.8	127.1 ± 42.3	114.0 92.5–165.8	0.13
Triglycerides [mg/dL]	98.0 ± 33.8	99.5 72.8–119.8	86.0 ± 38.2	76.0 54.8–115.3	0.20
Creatinine [mg/dL]	0.88 ± 0.16	0.85 0.80–0.98	0.86 ± 0.13	0.85 0.80–0.90	0.83
eGFR [mL/min/1.73 m^2^]	83.0 ± 9.9	82.9 77.1–90.7	83.8 ± 13.4	81.9 76.9–93.0	0.79
ACR [mg/g creatinine]	0.67± 0.63	0.45 0.20–0.88	1.28 ± 1.72	0.45 0.20–1.95	0.68

**Table 3 biomedicines-12-00863-t003:** Vascular parameters of the studied group. SBP_b_—systolic blood pressure measured in the brachial artery, DBP_b_—diastolic blood pressure measured in the brachial artery, MAP_b_—mean blood pressure measured in the brachial artery, PP_b_—pulse pressure measured in the brachial artery, HR_b_—heart rate measured in the brachial artery; SBP_c_—systolic blood pressure calculated in the aorta, DBP_c_—diastolic blood pressure calculated in the aorta, MAP_c_—mean blood pressure calculated in the aorta, PP_c_—pulse pressure measured calculated in the aorta, HR_c_—heart rate measured in the brachial artery during central blood pressure measurement; PWV—pulse wave velocity, APc—aortic augment pressure, Alx—augmentation index, IOP—intraocular pressure, OD—oculus dexter, OS—oculus sinister. SD—standard deviation; IQR—interquartile range between 25th percentile and 75th percentile.

Vascular Parameters	Control Group		DM 1		*p* Mann–Whitney U test
	Mean ± SD	Median IQR	Mean ± SD	Median IQR	
SBP_b_ [mmHg]	130.3 ± 10.6	131.0 124.5–137.0	132.4 ± 14.6	134.0 121.0–146.0	0.51
DBP_b_ [mmHg]	84.7 ± 8.5	85.0 78.5–90.0	83.7 ± 8.4	83.0 77.0–92.0	0.53
MAP_b_ [mmHg]	99.9 ± 8.6	101.7 94.7–105.5	100.0 ± 9.8	100.7 91.7–107.3	0.98
PP_b_ [mmHg]	45.6 ± 7.1	45.0 40.5–50.5	48.7 ± 9.8	49.0 42.0–55.0	0.15
HR_b_ [bpm]	66.1 ± 10.9	63.0 60.0–70.5	67.6 ± 9.8	65.0 62.0–72.0	0.40
SBP_c_ [mmHg]	118.2 ± 10.7	118.0 112.0–125.5	120.4 ± 13.5	122.0 110.0–133.0	0.38
DBP_c_ [mmHg]	85.5 ± 8.6	87.0 79.5–91.0	84.6 ± 8.6	83.0 79.0–92.0	0.59
MAP_c_ [mmHg]	98.5 ± 8.8	100.0 92.0–103.5	99.3 ± 9.9	100.0 92.0–106.0	0.87
PP_c_ [mmHg]	32.7 ± 7.5	31.0 27.0–37.5	35.7 ± 9.1	37.0 27.0–41.0	0.13
HR_c_ [bpm]	70.2 ± 10.8	70.0 62.5–78.5	72.2 ± 11.1	72.0 65.0–78.0	0.52
PWV [m/s]	6.1 ± 1.1	6.3 5.1–6.9	6.4 ± 1.4	6.4 5.1–7.6	0.45
APc [mmHg]	6.0 ± 6.8	5.0 1.0–9.0	8.26 ± 6.4	7.0 3.0–12.0	0.14
Alx [%]	15.9 ± 16.2	18.0 4.5–27.0	20.8 ± 14.1	20.0 10.0–32.0	0.22
IOP OD [mmHg]	14.0 ± 1.8	14.0 13.5–15.0	16.9 ± 3.2	17.0 15.3–18.8	0.004
IOP OS [mmHg]	14.1 ± 2.1	14.0 12.0–16.0	16.9 ± 3.4	17.5 14.5–19.0	0.006

**Table 4 biomedicines-12-00863-t004:** Retinal arteriolar morphological parameters. VD—vessel diameters, LD—lumen diameter, WT—wall thickness, WLR—wall-to-lumen ratio, WCSA—wall cross-sectional areas. SD—standard deviation; IQR—interquartile range between 25th percentile and 75th percentile.

Retinal Arteriolar Morphological Parameters	Control Group		DM 1		*p* Mann–Whitney U test
	Mean ± SD	Median IQR	Mean ± SD	Median IQR	
VD right eye [µm]	119.4 ± 13.5	119.9 112.0–127.4	123.1 ± 24.4	124.1 104.9–140.3	0.44
VD left eye [µm]	119.3 ± 12.8	120.1 109.4–128.5	123.4 ± 23.1	124.6 110.7–136.4	0.19
LD right eye [µm]	96.6 ± 12.7	96.0 87.7–102.3	96.0 ± 21.7	93.9 79.3–110.0	0.77
LD left eye [µm]	93.6 ± 17.2	96.5 87.4–103.3	95.8 ± 19.4	99.8 86.2–107.9	0.45
WT right eye [µm]	11.42 ± 1.62	11.45 10.48–12.10	13.57 ± 3.40	12.35 10.95–16.00	0.020
WT left eye [µm]	11.59 ± 1.36	11.45 10.61–12.29	13.76 ± 3.47	13.40 10.98–16.05	0.009
WLR right eye	0.24 ± 0.04	0.24 0.22–0.27	0.29 ± 0.09	0.30 0.22–0.34	0.018
WLR left eye	0.24 ± 0.03	0.24 0.22–0.27	0.29 ± 0.08	0.27 0.23–0.33	0.014
WCSA right eye [µm^2^]	3889 ± 824	38307 3391–4196	4764 ± 1847	4470 3531–5992	0.040
WCSA left eye [µm^2^]	3872 ± 852	3825 3313–4399	4757 ± 1878	4709 3665–5952	0.016

**Table 5 biomedicines-12-00863-t005:** Retinal cones morphological parameters. *n*—number of cones, D—cones density, S—spacing of cones, R—regularity of cones, DP—dispersion index of cones. SD—standard deviation; IQR—interquartile range between 25th percentile and 75th percentile.

Cones Morphological Parameters	Control Group		DM 1		*p* Mann–Whitney U test
	Mean ± SD	Median IQR	Mean ± SD	Median IQR	
*n* right eye	93.1 ± 9.3	93.5 85.5–99.8	91.9 ± 14.8	92.0 85.0–102.5	0.99
*n* left eye	95.1 ± 10.3	95.0 86.8–102.0	87.4 ± 10.2	90.0 81.0–94.0	0.016
D right eye [cells/mm^2^]	24,812 ± 2299	24,655 (22,932–26,663)	24,685 ± 3990	26,039 (21,937–27,971)	0.43
D left eye [cells/mm^2^]	25,510 ± 27,748	24,875 (23,208–28,145)	23,495 ± 3585	24,128 (21,291–25,208)	0.045
S right eye [µm]	7.01 ± 0.33	7.03 (6.73–7.30)	7.11 ± 0.67	6.79 (6.62–7.67)	0.59
S leftt eye [µm]	6.90 ± 0.43	6.97 (6.56–7.19)	7.18 ± 0.62	7.08 (6.94–7.61)	0.043
R right eye [%]	96.5 ± 2.4	96.8 (95.0–98.1)	94.0 ± 4.5	93.1 (91.0–98.5)	0.047
R left eye [%]	96.6 ± 2.5	97.0 (95.1–98.9)	95.0 ± 3.5	95.3 (92.4–97.8)	0.076
DP right eye [%]	9.40 ± 2.05	9.25 (7.83–10.50)	10.97 ± 3.00	11.70 (8.75–12.30)	0.020
DP left t eye [%]	9.28 ± 2.09	8.65 (7.90–10.43)	10.30 ± 2.45	10.20 (8.30–12.30)	0.12

**Table 6 biomedicines-12-00863-t006:** Common carotid artery morphological parameters. IMT—intima/media thickness of the common carotid artery, LDCA—lumen diameter of the common carotid artery, IMTLR—intima/media thickness lumen ratio calculated by IMT/LDCA in both common carotid artery and averaged for calculation of the mean value. SD—standard deviation; IQR—interquartile range between 25th percentile and 75th percentile.

Common Carotid Artery Morphological Parameters	Control Group		DM 1		*p* Mann–Whitney U test
	Mean ± SD	Median IQR	Mean ± SD	Median IQR	
IMT right side [cm]	0.056 ± 0.015	0.055 0.050–0.070	0.072 ± 0.017	0.080 0.060–0.090	<0.001
IMT left side [cm]	0.055 ± 0.012	0.050 0.049–0.060	0.070 ± 0.015	0.070 0.060–0.080	<0.001
LDCA right side [cm]	0.55 ± 0.07	0.53 0.51–0.60	0.57 ± 0.09	0.56 0.51–0.61	0.34
LDCA left side [cm]	0.53 ± 0.06	0.53 0.50–0.56	0.56 ± 0.09	0.53 0.49–0.63	0.36
IMTLR right side [cm]	0.10 ± 0.02	0.11 0.08–0.12	0.13 ± 0.03	0.13 0.10–0.16	0.004
IMTLR left side [cm]	0.10 ± 0.02	0.10 0.09–0.11	0.13 ± 0.03	0.13 0.10–0.15	0.001

## Data Availability

The data that support the findings of this study are available from the corresponding author, upon reasonable request.

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
