# Peer review of "High-Resolution Imaging of Cones and Retinal Arteries in Patients with Diabetes Mellitus Type 1 Using Adaptive Optics (rtx1)"

_biomedicines, 2024, doi:10.3390/biomedicines12040863_

Round 1

Reviewer 1 Report

Comments and Suggestions for Authors

The manuscript presents a classic and straightforward biostatistics study applied to diabetic retinopathy. The methods seem correct, aside from some points indicated below, and the results and conclusion seem somewhat expected, so the authors might want to discuss a little more about the relevance of the manuscript.

Regarding formatting issues, I`ve detected the following:

- throughout the paper, the units shall be separated from the values (e.g. "13 %" instead of "13%", "420 nm" instead of "420nm", etc);

- throughout the paper, the thousands separator should be removed (e.g. "199 000" or "199000" instead of "199,000", as it creates ambiguity;

- the square exponents in mm2 and m2 shall be raised;

- table 2 presents the DM group first, and then the Control group, but the next Tables invert the groups;

- line 195 seems to be truncated;

In terms of scientific content, I have the following questions:

- in section 2.4, the authors declare that the t-test was used to compare MEDIAN values for data with normal distribution, and this seems odd because, in such cases, one would normally compare the MEAN values. Why medians?

- when describing the sample patients in Section 3, what does "no DR features" mean? This needs further clarification.

- in lines 220-221, what does a "tendency to correlate" mean? Does it mean that the p value was ALMOST below 0.05? This seems a little overeaching, especially not seeing the scatter plots, as the correlation values are actually quite small (<< 1.0)

Comments on the Quality of English Language

The English language quality is fine, with just a few typos that could be easily detected and corrected.

Author Response

Reviewer 1.

Dear Sir or Madame,

Thank you for the insightful and thorough review of our work. We appreciate your effort involved in reviewing such an extensive text. We considered all your comments and we hope to have resolved all the issues you suggested and correct the mistakes.

Below are the answers to your questions:

- throughout the paper, the units shall be separated from the values (e.g. "13 %" instead of "13%", "420 nm" instead of "420nm", etc); - Corrected

- throughout the paper, the thousands separator should be removed (e.g. "199 000" or "199000" instead of "199,000", as it creates ambiguity;- Corrected

- the square exponents in mm2 and m2 shall be raised; - Corrected

- table 2 presents the DM group first, and then the Control group, but the next Tables invert the groups; - Corrected

- line 195 seems to be truncated; - Corrected                   

In terms of scientific content, I have the following questions:

- in section 2.4, the authors declare that the t-test was used to compare MEDIAN values for data with normal distribution, and this seems odd because, in such cases, one would normally compare the MEAN values. Why medians?

We adapted the description of the statistics to our analyses. The analysis of normal distributions was not included because we did not have any. We have left only the description of non-parametric tests.

- in lines 220-221, what does a "tendency to correlate" mean? Does it mean that the p value was ALMOST below 0.05? This seems a little overeaching, especially not seeing the scatter plots, as the correlation values are actually quite small (<< 1.0)

I was thinking about a positive correlation that does not meet the criteria for statistical significance. But now I think it is appropriate to remove this part. Corrected.

- when describing the sample patients in Section 3, what does "no DR features" mean? This needs further clarification. 

I wanted to emphasize that these are patients without diabetic retinopathy, so I corrected on: without DR.

Thanks to your tips, in our opinion, the work has gained a new quality and value. We hope the revised manuscript now meets your and your readers' expectations.

Ps. Following the suggestion of one of the reviewers, we changed the title.

Yours faithfully,

Authors

Reviewer 2 Report

Comments and Suggestions for Authors

The aim of this manuscript is to assess retinal cones in DM1 patients without previously diagnosed DR compared to a group of  healthy volunteers. . Overall, I think the concept behind the study is of interest. In fact, I don't have much to add to the study design or the construction of the paper. But there are still some minor grammar error needed to edit. Please have some native English speaker revise it to make it easier to read.

Specifically, the authors should fix the following problems which will help the work achieve the published criteria of this journal.

1. In lins 89, it is much better to use ocular assessment other than eye assessment.

2. Please refer to a statistician to evaluate the statistical methods used in the text. Since both eyes were enrolled, the method used in the text is not suitable.

3. It is wierd the number of cones is correlated with blood pressure. Why?

4. Additionally, the number of cones in DM1 with hypertension is lower than without hypertension. However, there was not statistically difference in the control group. Is it by chance?

Comments on the Quality of English Language

The aim of this manuscript is to assess retinal cones in DM1 patients without previously diagnosed DR compared to a group of  healthy volunteers. . Overall, I think the concept behind the study is of interest. In fact, I don't have much to add to the study design or the construction of the paper. But there are still some minor grammar error needed to edit. Please have some native English speaker revise it to make it easier to read.

Specifically, the authors should fix the following problems which will help the work achieve the published criteria of this journal.

1. In lins 89, it is much better to use ocular assessment other than eye assessment.

2. Please refer to a statistician to evaluate the statistical methods used in the text. Since both eyes were enrolled, the method used in the text is not suitable.

3. It is wierd the number of cones is correlated with blood pressure. Why?

4. Additionally, the number of cones in DM1 with hypertension is lower than without hypertension. However, there was not statistically difference in the control group. Is it by chance?

Author Response

Reviewer 2.

Dear Sir or Madame,

Thank you for the insightful and thorough review of our work. We appreciate your effort involved in reviewing such an extensive text. We considered all your comments and we hope to have resolved all the issues you suggested and correct the mistakes.

Below are the answers to your questions:

  1. In lins 89, it is much better to use ocular assessment other than eye assessment. Corrected
  2. Please refer to a statistician to evaluate the statistical methods used in the text. Since both eyes were enrolled, the method used in the text is not suitable.

Morphological parameters of retinal cones and vessels  were measured with the AO rtx1 camera in right and left eyes. The values determined for both eyes were averaged and mean values were calculated. Corrected.

  1. It is wierd the number of cones is correlated with blood pressure. Why?

Hypertension causes vascular damage (remodeling in hypertension arterial). Diabetes causes non-enzymatic glycation in vessels, e.g. repair enzymes, which causes the situation in the vessels to worsen and microangiopathies. What causes deterioration of nutrient supply in

tissue. Diabetes causes diabetic retinopathy, in this work we observe this as the disappearance of cones. This loss of cones and abnormalities in their structure may be very early markers of diabetic retinopathy. Below is an excellent article on a similar topic.

van Varik BJ, Rennenberg RJ, Reutelingsperger CP, Kroon AA, de Leeuw PW, Schurgers LJ. Mechanisms of arterial remodeling: lessons from genetic diseases. Front Genet. 2012 Dec 13;3:290. doi: 10.3389/fgene.2012.00290. PMID: 23248645; PMCID: PMC3521155.

  1. Additionally, the number of cones in DM1 with hypertension is lower than without hypertension. However, there was not statistically difference in the control group. Is it by chance?

We think that, along with hyperglycemia and non-enzymatic glycation, hypertension is a significant risk factor for the development of diabetic retinopathy. Cone atrophy is very early symptom of DR seen in AO. That is why the number of cones is lower in the group with hypertension, perhaps we could obtain statistical significance by studying a larger group of patients.  

Thanks to your tips, in our opinion, the work has gained a new quality and value. We hope the revised manuscript now meets your and your readers' expectations.

Ps. Following the suggestion of one of the reviewers, we changed the title.

Yours faithfully,

Authors

Reviewer 3 Report

Comments and Suggestions for Authors

In the current manuscript entitled “Assessment of the retina with focus on cones in type 1 diabetes mellitus patients with the use of adaptive optics", Wojciech Matuszewski and colleagues aimed to study retinal cones in patients with a history of type 1 diabetes without previously diagnosed DR compared to healthy volunteers to identity changes in the number or morphology of cones as an indicator for later developing symptoms of DR.

The overall outcomes of the manuscript are relevant and represent a potent translation application since it provides a method to perform non-invasive in vivo assessment of retinal cones.

One of the limitations of the manuscript relies on the lack of patients with a history of type 2 diabetes and that must be addressed/discussed in the manuscript considering that among the poll of patients with a history of diabetes ~90 % suffer of type 2 DM. Whether T2D differs in the etiology of changes in the morphology of the cones, or the authors decided to focus only in T1D, should be included in the rational, retrospective design and in the discussion.

The title should be rewritten using proper English.

The length of the manuscript and especially the discussion should be trimmed about 1/3.

In conclusion, authors should sharply focus on their message and avoid repeating what has already been reviewed in the discussion. 

Comments on the Quality of English Language

-

Author Response

Reviewer 3.

Dear Sir or Madame,

Thank you for the insightful and thorough review of our work. We appreciate your effort involved in reviewing such an extensive text. We considered all your comments and we hope to have resolved all the issues you suggested and correct the mistakes.

Below are the answers to your questions:

- One of the limitations of the manuscript relies on the lack of patients with a history of type 2 diabetes and that must be addressed/discussed in the manuscript considering that among the poll of patients with a history of diabetes ~90 % suffer of type 2 DM. Whether T2D differs in the etiology of changes in the morphology of the cones, or the authors decided to focus only in T1D, should be included in the rational, retrospective design and in the discussion. 

Of course I agree that the vast majority of diabetes is DM2. However, I am very interested and focus in my work on patients with DM1. Currently, we are also observing an increase in the incidence of DM1. In the discussion, I referred to interesting but unfortunately few similar works on DM1. Therefore, we also compared our results to articles about DM 2. In the next work, we plan to compare the changes in cones and retinal vessels in the DM1 and DM2 groups.

- The title should be rewritten using proper English.  New title: Adaptive Optics (rtx1) High-Resolution Imaging of Cones and Retinal Arteries in Patients with Diabetes Mellitus type 1.

- The length of the manuscript and especially the discussion should be trimmed about 1/3.

Corrected, I removed the beginning of the discussion, which was a repetition of information from the results. 

- In conclusion, authors should sharply focus on their message and avoid repeating what has already been reviewed in the discussion. 

Corrected. I shortened the conclusions and left only the important information.

Thanks to your tips, in our opinion, the work has gained a new quality and value. We hope the revised manuscript now meets your and your readers' expectations.

Yours faithfully,

Authors

Round 2

Reviewer 2 Report

Comments and Suggestions for Authors

It is not suitable to average the values of right and left eye.  Please use  Generalized estimating equation (GEE) models  to count for inter-correlation of eyes within study subjects. Eyes (left or right) should set as within-subject variables in the GEE models.

Comments on the Quality of English Language

The English writing can be improved.

Author Response

Reviewer 2.

Dear Sir or Madame,

Thank you for your valid comments, we have made changes to the tables and assessed the right and left eyes separately. We did this because we have little experience with GEE, but we hope you will accept our changes. Currently, the work is more detailed and I hope it will be appreciated by readers.

Yours faithfully,

Authors
